# Influence of Acid-Modified Clinoptilolite on the Self-Adhesive Properties of Silicone Pressure-Sensitive Adhesives

**DOI:** 10.3390/polym15030707

**Published:** 2023-01-31

**Authors:** Adrian Krzysztof Antosik, Marlena Musik, Piotr Miądlicki, Mateusz Weisbrodt, Katarzyna Wilpiszewska

**Affiliations:** 1Department of Chemical Organic Technology and Polymeric Materials, Faculty of Chemical Technology and Engineering, West Pomeranian University of Technology in Szczecin, Pułaskiego 10, 70-322 Szczecin, Poland; 2Department of Engineering of Catalytic and Sorbent Materials, Faculty of Chemical Technology and Engineering, West Pomeranian University of Technology in Szczecin, Piastów Ave. 42, 71-065 Szczecin, Poland

**Keywords:** acid modification, clinoptilolite, silicone pressure-sensitive adhesive, self-adhesive properties, adhesion

## Abstract

The preparation of a new “eternally alive adhesive” based on silicone pressure-sensitive adhesives with clinoptilolite is presented. Neat and acid-modified (i.e., treated with sulfuric acid (VI)) clinoptilolite was used. The effect of clinoptilolite acid treatment on the adhesive properties of pressure-sensitive adhesive tapes was tested. The obtained tapes exhibited increased thermal resistance when compared to the reference tapes. Despite introducing the filler, the pressure-sensitive adhesive tapes maintained good functional properties. The new self-adhesive materials show promising implementation potential where increased thermal resistance is required.

## 1. Introduction

A primitive concept of adhesive tape, based on a starch-based paste applied on strips of cloth, was used by the ancient Egyptians [1,2,3]. However, the long history of adhesive materials, self-adhesive tape technology is a fairly new concept, originating in the mainstream history of adhesives as one of its youngest branches. 

The pressure-sensitive adhesives (PSAs) are defined as a group of adhesives exhibiting significant adhesive strength and tack at room temperature. Importantly, good adhesive parameters do not result from the chemical reaction between the adhesive and the substrate [2,3,4]. The most important performance properties defining PSAs are: adhesion, tack, and cohesion. These determine the adhesive–cohesive balance between the internal cohesion of the adhesive film and its interaction with the substrate [5,6,7,8]. PSAs play an important role in everyday life and have been intensively developed over the last decade. The most common are solvent-borne PSAs [9,10,11]. These PSAs are used for connecting various materials, such as metal, paper, plastics, glass, wood, or leather. They are applied for use in labels, protective films, mounting tapes, masking tapes, advertising banners, and medical products (including patches, bandages, surgical tapes, and biomedical electrodes). These high-quality self-adhesives should exhibit a constant level of shear strength, as well as excellent resistance to aging (at room and elevated temperatures). Moreover, they should be resistant to light, oxygen, moisture, and other environmental factors, such as the high daily amplitude of temperature [2,12,13,14,15,16]. The adhesive layer must be able to be applied on a flexible carrier (fabric, foil, paper), while maintaining a very long tack life (however, protection against dirt is required). The polymer liquid is applied on the carrier by rollers and subsequently, within a few seconds, the solvent evaporation in a drying channel and the cross-linking of the polymer chains are performed.

Among the pressure-sensitive adhesives, silicone pressure-sensitive adhesives (Si-PSAs) are considered as materials for special applications. They can be applied in a wide temperature range, from −40 up to 300 °C, on low- and high-energy surfaces. Si-PSAs are flexible, inert, hydrophobic, non-toxic, and biocompatible. The molecular weight of the solvent silicones ranges from 500,000 to 1,500,000 Daltons. The cross-linking of silicone PSAs is usually carried out at 120–150 °C using organic peroxides, such as benzoyl peroxide (BPO) and dichlorobenzoyl peroxide (DClBPO), due to their small number of functional groups—methyl and/or phenyl. In recent years, the additional curing of silicone PSAs was reported [12,17,18]. Si-PSAs have been commercially produced since the 1960s, and are applied in heavy industry, electrical and electronic engineering, healthcare, and the automotive industry. Since 2000, the growing interest in the medical applications of these PSAs has been noted. Silicone pressure-sensitive adhesives are used when application conditions or the nature of the substrate surface exceed the performance limits of organic PSAs. The main structure of the silicone adhesive consists of a polysiloxane skeleton, i.e., silicon-oxygen bonds, and the bond energy is much higher than the C-C bond. This results in unique thermal resistance and stability. Moreover, silicone PSAs exhibit high chain flexibility. The methyl groups attached to the Si atoms are sterically undisturbed and can rotate freely; thus, the main polar chain -Si-O-Si- is shielded and in consequence, the intra- and intermolecular interactions are minimized. Therefore, to improve its mechanical properties, high-density cross-linking is necessary [19,20,21,22] Driven by the need for regulatory compliance and changing performance requirements, silicone PSAs, based on new silicone chemicals and curing mechanisms, have emerged [18,23,24].

The acid modification of zeolites is commonly used to increase their ion-exchange capacity and to obtain a more selective material [25,26,27].

Rakitskaya et al. determined the effect of nitric acid on clinoptilolite’s physicochemical and structural properties. Moreover, the catalytic activity of modified clinoptilolite complexes (anchored in palladium and copper) in the low-temperature oxidation reaction of carbon monoxide was investigated [28].

Inorganic and organic acids were used for the surface modification of clinoptilolite by Kadirbekov et al. Treatment with 10% H_4_EDTA/Kl (a natural zeolite modified with ethylene diamine tetraacetic acid) increased the specific surface area from 9.8 m^2^ g^−1^ up to 28.2 m^2^ g^−1^, and treatment with HKl-1 (a zeolite subjected to a single modification by an inorganic acid) increased the area by 52.6 m^2^ g^−1^. Moreover, the gradual modification (first with inorganic acids and then with organic acids or heteropolymers) resulted in additional increases in specific surface area up to 99.4 m^2^ g^−1^ for 10% H_4_EDTA/HKl-1 (a zeolite decationized with an inorganic acid and modified with ethylene diamine tetraacetic acid) and 257.0 m^2^ g^−1^ for 10% PW_12_-HPA/HKl-1 (a zeolite decationized with an inorganic acid and modified with tungsten heteropoly acid), respectively. The amount and pore volume of the clinoptilolite increased when the surface was acid-activated [29].

Miądlicki et al. modified clinoptilolite with various concentrations of H_2_SO_4_ solutions. It was used as an effective “green catalyst” for the isomerization of α-pinene. Moreover, the clinoptilolite modification process did not require complex or expensive apparatus and was relatively easy to perform on a large scale [30].

In this paper, the preparation of new self-adhesive pressure-sensitive adhesives (based on commercial silicones) with clinoptilolite is described. Subsequently, the obtained adhesives were used for the preparation of one-side self-adhesive tapes. Neat and acid-modified clinoptilolite was applied. The effect of filler modification on the self-adhesive properties (adhesion, cohesion, stickiness, skinning) was investigated. The pot life of the adhesive was determined. Its thermal resistance was tested using SAFT tests (shear adhesive failure temperature).

## 2. Materials and Methods

### 2.1. Materials

Two commercial silicone resins, Q2-7358 and Q2-7355, were purchased from Dow Corning (Midland, MI, USA). Dichlorobenzoyl peroxide—DClBPO Gelest (Morrisville, PA, USA) was used as the crosslinker. Toluene was purchased from Carl Roth (Karlsruhe, Germany), and clinoptilolite was obtained from Turkey. Sulfuric acid 95% was purchased from POCH (Poland).

### 2.2. Methods

#### 2.2.1. Filler Modification

Clinoptilolite was modified with aqueous solutions of sulfuric acid in a concentration from 0.01 up to 2 M (10 cm^3^ of the corresponding acid solution per 1 g of clinoptilolite). The aqueous suspension was mixed with a mechanical stirrer (500 rpm) for 4 h at 80 °C. The modified clinoptilolite was filtered and washed with distilled water until no SO_4_^2−^ ions were detected in the filtrate. The samples were dried at 100 °C for 24 h. The acronyms of the natural and modified clinoptilolite are shown in Table 1.

#### 2.2.2. Preparation of One-Side Si-PSA Tapes

The silicone resin was mixed with the filler and a crosslinker (1.5 wt% based on the resin weight) in toluene to obtain a homogenous composition containing 50 wt% polymer. The filler content was: 0.1, 0.5, 1, and 3 wt% (based on the resin). The composition was coated onto a polyester film (50 g/m^2^) using a semi-automatic coater and was subsequently placed in a drying duct for 10 min at 110 °C for cross-linking. The obtained adhesive film was secured with a polyester film.

#### 2.2.3. X-ray Diffraction (XRD)

X-ray diffraction (XRD) analysis was carried out to determine the crystal structure of the modified clinoptilolite. The XRD patterns were recorded by an Empyrean PANalytical X-ray diffractometer (Malvern Panalytical Ltd, Malvern, UK), with a Cu lamp used as the radiation source in the 2θ 9–34° range with a step size of 0.026.

#### 2.2.4. Fourier Transform Infrared Spectroscopy (FT-IR)

For each sample, FT-IR spectra were obtained with a Thermo Nicolet 380 (Waltham, MA, USA) spectrometer with ATR unit in the range of wavenumbers from 400 to 4000 cm^−1^.

#### 2.2.5. Pot Life

The pot life was defined as the time after which the viscosity increased (two or four times) when compared to the viscosity of the fresh mixture. The tests were carried out at room temperature immediately after mixing the adhesive components [31].

#### 2.2.6. Tack

Tack indicates the ability of the adhesive to bond briefly (without pressure) with the given surface. It can also be defined as the force needed to separate the surfaces after a short time. The tests were carried out according to the technical standard FINAT FTM9 [32].

#### 2.2.7. Adhesion

Adhesion is defined as the interaction of the surfaces of two bodies or phases. It is closely related to the forces of interfacial tension on the contact surfaces of both materials. The work test was performed according to the FINAT FTM 1 method [33].

#### 2.2.8. Cohesion

Cohesion refers to the strength of the adhesive joint. Next to adhesion, it is one of the most important properties of adhesives. Temperature, type of cross-linking compounds, or thickness of the adhesive film affect the cohesion value. Cohesion was determined by the FINAT FTM 8 method. The measurement was carried out at room and elevated temperature (70 °C) [34,35].

#### 2.2.9. Shrinkage

Shrinkage is defined as the reduction in the surface area of the adhesive when compared to its initial size. It is an important mechanical and functional property, especially when it can result in surface deformations. In the case of pressure-sensitive adhesives, the change in surface area is given in millimeters or percentages. The value above 0.5 mm or 0.5% exceeds the acceptable shrinkage in the technology of self-adhesive products [36].

#### 2.2.10. SAFT Test

The resistance to elevated temperature was tested using the SAFT test. The measurements were performed in a similar way to those for cohesion; however, the temperature during the test increased up to 230 °C (2 °C/min) [37].

The detailed descriptions of the research methods regarding these properties (adhesion, cohesion, and tack) have been reported in previous papers [38,39].

## 3. Results and Discussion

### 3.1. Clinoptilolite Characterization

To enhance the compatibility between the hydrophilic filler and the hydrophobic polymer matrix, clinoptilolite was modified with sulfuric acid. This process affects the chemical composition of the filler, its roughness, and its surface energy [40] The FTIR spectra of neat and acid-modified clinoptilolite were shown in Figure 1. The broad bands between 2900 and 3745 cm^−1^ could be attributed to the O-H stretching vibration mode of water adsorbed in the zeolite (water molecules bound to Na and Ca in the channels and cages of the zeolite structure), intermolecular hydrogen bonds, and Si-OH-Al bridges [30]. The characteristic band at 1630 cm^−1^ was ascribed to the H_2_O bending vibration. The band detected at 600 cm^−1^ was attributed to bending vibrations between tetrahedra, specifically, double ring vibrations [41]. The band at 787 cm^−1^ belonged to the Si-O-Si bonds [42]. The band at 445 cm^−1^ was characterized by pore opening (O-T-O bending vibrations, where T = Al, Si). The strong band at 1020 cm^−1^ was attributed to Si-O-Si, which can be overlapped with Al-O-Si and Al-O stretching vibrations. The position of this band depends on the Al/Si ratio and indicates the number of Al atoms per unit [41]. The shifting of this band to a higher wave number was observed in the spectra of modified clinoptilolite and could be associated with an increase in the Si/Al ratio in the clinoptilolite backbone after acid modification. Moreover, for clinoptilolite etched with 0.5 M, 1 M, and 2 M acid solutions, the increased intensity of this band indicated the Si/Al ratio increase. The bands at 725, 669, 600, and 520 cm^−1^ were attributed to the off-frame cations in the clinoptilolite matrix [42,43]. The off-framework cations were completely removed using concentrated sulfuric acid solutions (1 M and higher).

The XRD diffractograms of neat and modified clinoptilolite are presented in Figure 2. The peaks characteristic for clinoptilolite, according to JCPDS sheet 25-1349 (2θ = 9.85°, 11.19°, 13.09°, 16.92°, 17.31°, 19.09°, 20.42°, 22.48°, 22.75°, 25.06°, 26.05°, 28.02°, 28.58°, 29.07°, 30.12°, 31.97°, and 32.77°), were observed in the unmodified and acid-treated clinoptilolite. The signals at 20.86 and 26.60° were assigned to quartz (according to JCPDS 85-0930 card 85-0930). The intensity of the peaks corresponding to the clinoptilolite at ca. 22.5° decreased with the acid concentration of the solution used for modification. This was associated with some structural changes and the lowering of the clinoptilolite’s crystallinity. Quartz is inert to sulfuric acid; therefore, the intensities of the quartz peaks in the spectrum were similar for all samples.

### 3.2. Silicone Pressure-Sensitive Adhesives

In the next step, acid-modified clinoptilolite was used as a component of silicone pressure-sensitive adhesives. Two commercial silicone resins were selected: Q2-7358 and Q2-7355. The physical properties (adhesion, cohesion, and tack) of the systems without the filler are shown in Table 2. The adhesive films exhibited good functional properties, with notably excellent cohesion.

The effect of clinoptilolite content on the viscosity of silicone pressure-sensitive adhesive compositions is presented in Table 3 and Table 4. The lowest viscosity values were noted for the neat systems. The viscosity values increased with time and filler content. Moreover, the adhesives containing filler modified with a higher concentration of acid solution exhibited higher viscosity when compared to the systems with the same filler load. This is likely caused by the increase in the number of silicon groups. In the case of the Q2-7358 resin-based compositions, all the systems exhibited gelation after 7 days; therefore, coating them was no longer possible. For the Q2-7355 resin-based systems, some of the samples exhibited gelation on the second day. Thus, such adhesive compositions could not be stored on a long-term basis, requiring real-time coating upon receipt. This tendency was reported for many adhesives containing mineral fillers [44,45].

The effect of clinoptilolite content on peel adhesion and tack is presented in Figure 3 and Figure 4 and Figure 5 and Figure 6, respectively. Generally, the adhesion of the systems based on the Q2-7358 resin decreased after the filler addition. In the case of the adhesives based on Q2-7355 resin, the adhesion was improved in the presence of a modified filler, and generally, the higher the acid concentration for filler modification, the higher the adhesion. This was probably due to the better ordering and tighter structure of the adhesive film [46,47]. The highest values of adhesion, about 13 and 11.5 N/25mm for Q2-7358 and Q2-7355, respectively, were noted for the systems containing clinoptilolite modified with the highest concentration of the acid solution. The maximum tack value (ca. 11.5 N) was noted for the Q2-7358 adhesives containing clinoptilolite etched with the 0.5 M acid solution, whereas for the Q2-7355 adhesive, the maximum tack value (ca. 9.5 N) was noted for the system with filler etched with the 2 M acid solution.

In Table 5 and Table 6, the cohesion values at room and elevated temperature are given. For the Q2-7355 resin-based systems, nearly every composition exhibited a higher cohesion value than that required by the self-adhesive tapes production industry (above 72 h). Specifically, only in the case of the Q2-7358 resin-based adhesives did the filler content caused a decrease in cohesion value below the required standard (0.1 and 3.0 wt%).

The SAFT (shear adhesion failure temperature) test results are shown in Table 7. In most cases, the adhesives with the fillers exhibited a very good thermal resistance (exceeding those of the reference systems). For both resins, the test limit (>225 °C) was reached by three samples, i.e., those containing 0.1 wt% of neat filler and 0.1 wt% clinoptilolite etched with 0.01 to 0.5 M acid solutions.

The shrinkage values of the obtained adhesives are presented in Table 8 and Table 9. Generally, the value of this parameter decreased with clinoptilolite content. This is due to the better alignment of the polymer net and the more compact internal structure of the adhesive film [48,49,50]. For the Q2-7358 resin-based systems, the 0.1 wt% filler content appeared to be too low to achieve the required shrinkage, i.e., below 0.5%. In the case of the Q2-7355 resin-based adhesives, a similar phenomenon was noted for 0.1 and 0.5 wt% filler content. The systems with a higher filler load exhibited a lower shrinkage than required. Interestingly, the acid-modified clinoptilolite slightly modified the shrinkage value.

## 4. Conclusions

New self-adhesive silicone adhesives including clinoptilolite were obtained. The filler was chemically modified by acid etching with sulfuric acid (VI). The prepared one-side self-adhesive tapes exhibited good functional properties, such as adhesion, cohesion, and tack. More importantly, the thermal resistance was improved when compared to that of the reference tapes.

Such materials could be used for covering installations and fireplaces, for connecting elements exposed to high temperatures in households, in hot-stamping technology, and in heavy and automotive industries. These materials also have applications in aeronautics, i.e., as a binder for solar batteries on satellite decks, and could be used in masking tapes in powder coating processes.

## Figures and Tables

**Figure 1 polymers-15-00707-f001:**
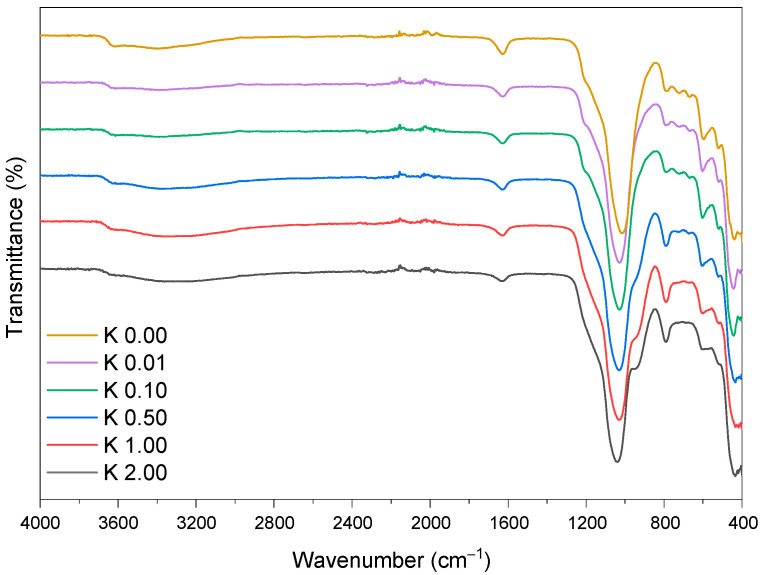
FTIR spectra of the neat and modified clinoptilolite.

**Figure 2 polymers-15-00707-f002:**
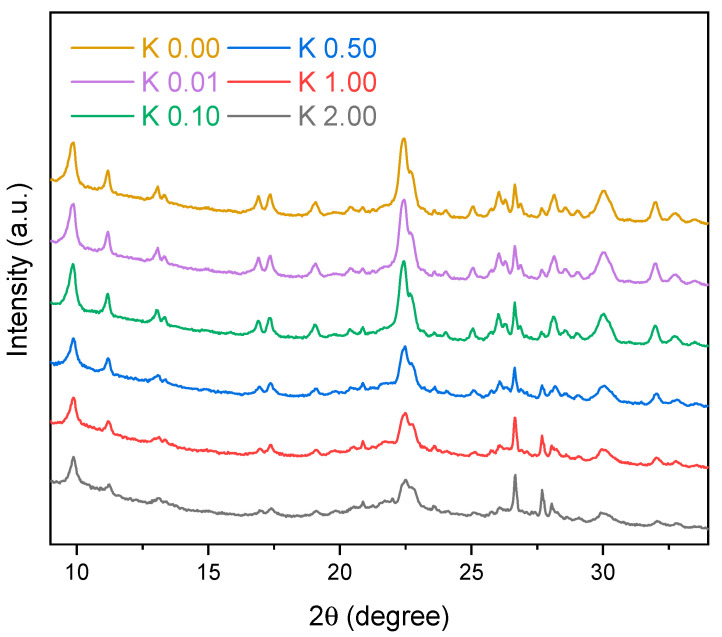
XRD diffractograms of neat and modified clinoptilolite.

**Figure 3 polymers-15-00707-f003:**
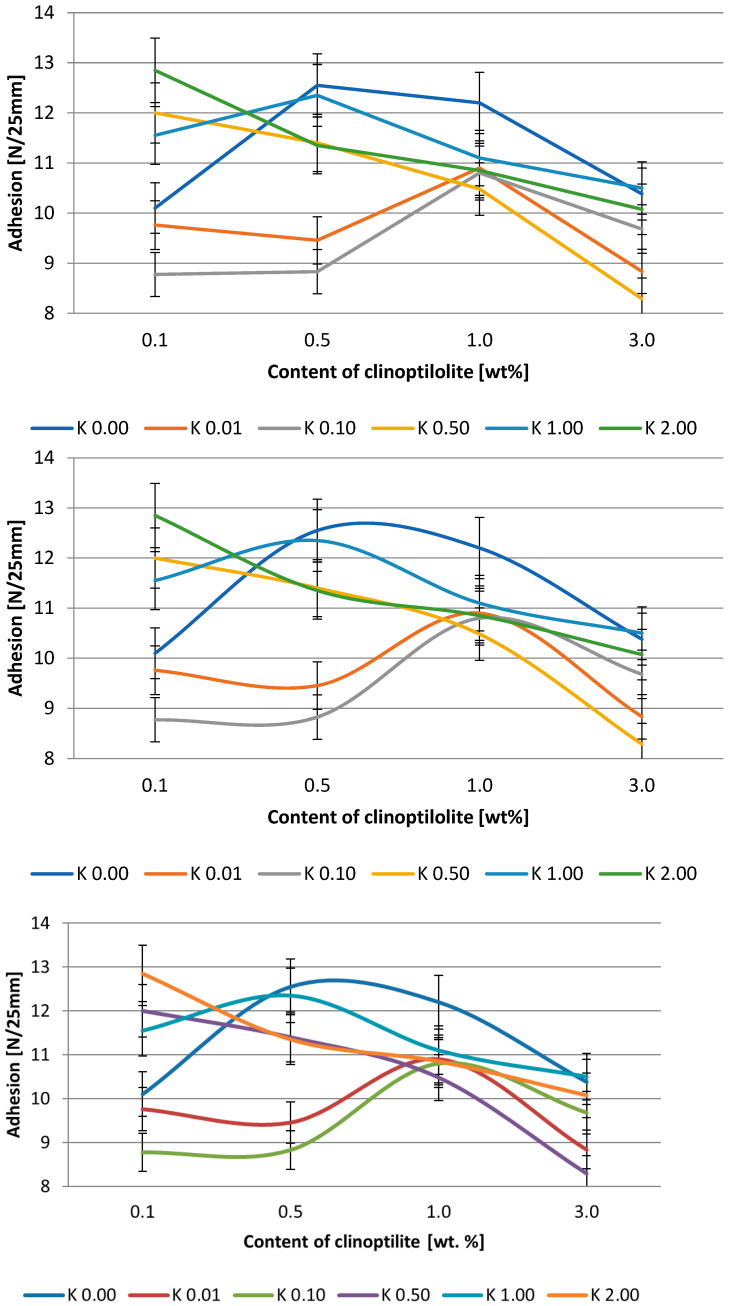
Effect of clinoptilolite content on the adhesion of silicone pressure-sensitive adhesives based on Q2-7358 resin.

**Figure 4 polymers-15-00707-f004:**
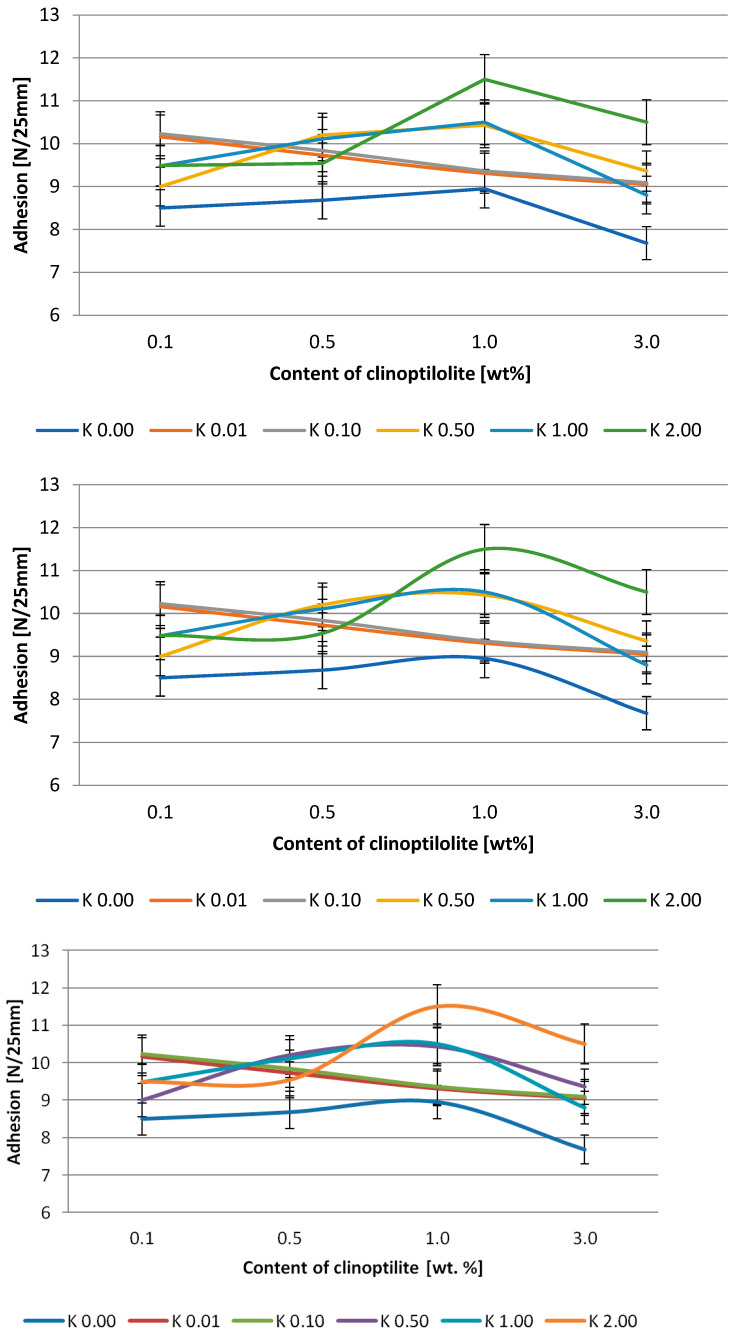
Effect of clinoptilolite content on the adhesion of silicone pressure-sensitive adhesives based on Q2-7355 resin.

**Figure 5 polymers-15-00707-f005:**
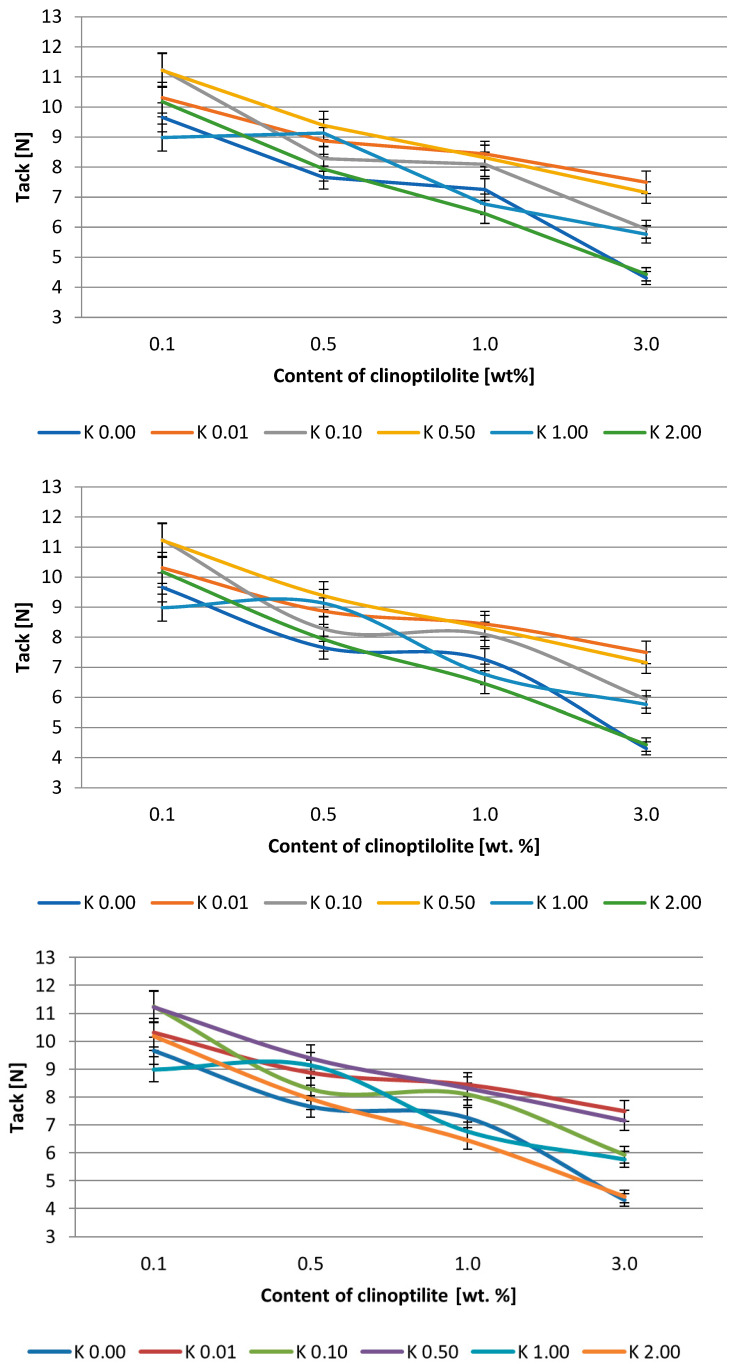
Effect of clinoptilolite content on the tack of silicone pressure-sensitive adhesives based on Q2-7358 resin.

**Figure 6 polymers-15-00707-f006:**
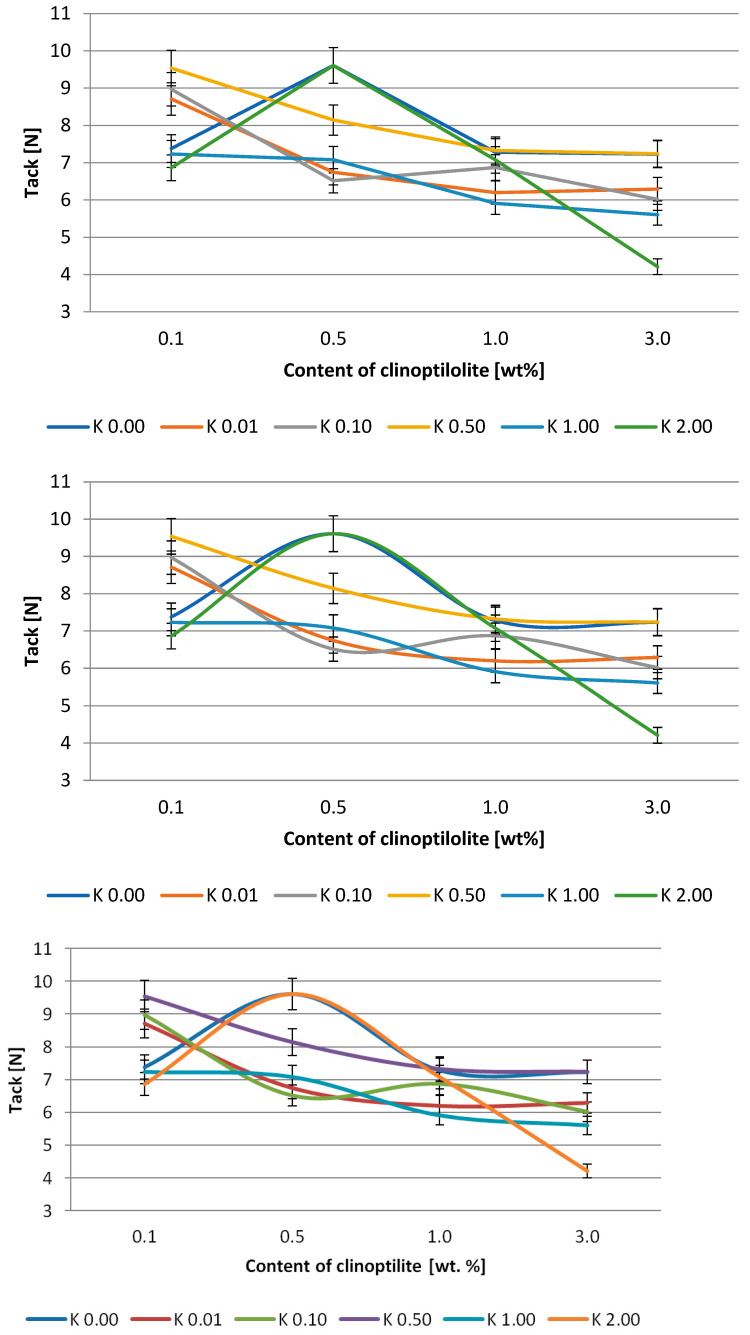
Effect of clinoptilolite content on the tack of silicone pressure-sensitive adhesives based on Q2-7355 resin.

**Table 1 polymers-15-00707-t001:** Acronyms of natural and modified clinoptilolite.

Filler	Treatment	Acronym
Clinoptilolite	Natural, unmodified	K 0.00
	0.01 M H_2_SO_4_	K 0.01
	0.1 M H_2_SO_4_	K 0.10
	0.5 M H_2_SO_4_	K 0.50
	1.0 M H_2_SO_4_	K 1.00
	2.0 M H_2_SO_4_	K 2.00

**Table 2 polymers-15-00707-t002:** Physical properties of one-side Si-PSA tape prepared without the filler.

Resin Acronym	Tack [N]	Adhesion [N/25 mm]	Cohesion [h]		SAFT [°C]
			20 °C	70 °C	
Q2-7358	6.9	10.2	>72	>72	147
Q2-7355	4.8	7.2	>72	>72	158

**Table 3 polymers-15-00707-t003:** The viscosity of Q2-7358 resin-based compositions with clinoptilolite.

Filler Content [wt%]	Viscosity [Pa·s]					
	K 0.00	K 0.01	K 0.10	K 0.50	K 1.00	K 2.00
After 1 day						
0.1	15.2	15.3	15.2	15.8	15.4	15.8
0.5	15.2	16.5	16.6	15.4	15.8	16.4
1.0	15.8	16.3	16.1	16.1	16.4	17.0
3.0	16.9	16.9	16.8	16.8	17.1	18.2
After 2 days						
0.1	16.5	16.5	16.4	16.5	16.2	16.7
0.5	16.6	17.4	17.9	16.8	16.7	17.5
1.0	16.6	17.1	17.0	16.7	18	18.2
3.0	17.1	17.6	18.4	18.0	18.6	18.3
After 7 days						
0.1	29.4	29.8	28.8	28.8	28.4	27.3
0.5	30.7	31.4	30.8	29.9	30.3	29.9
1.0	31.6	31.8	31.6	29.2	31.8	32.0
3.0	30.7	31.4	35.0	30.8	35.3	33.7

**Table 4 polymers-15-00707-t004:** The viscosity of Q2-7355 resin-based compositions with clinoptilolite.

Filler Content [wt%]	Viscosity [Pa∙s]					
	K 0.00	K 0.01	K 0.10	K 0.50	K 1.00	K 2.00
After 1 day						
0.1	2.1	2.3	1.8	8.0	2.8	1.2
0.5	5.7	6.2	5.0	22.1	7.8	3.4
1.0	4.9	5.3	4.3	18.9	6.7	2.9
3.0	4.8	5.2	4.2	18.6	6.6	2.8
After 2 days						
0.1	31.1	35.6	28.0	35.3	47.6	8.5
0.5	34.1	39.1	30.7	38.7	52.1	9.3
1.0	30.5	34.6	27.2	34.3	46.1	8.2
3.0	29.1	33.4	26.2	33.1	44.5	7.9
After 7 days						
0.1	615.5	728.0	656.6	705.3	643.1	519.8
0.5	593.5	702.0	633.2	680.2	620.2	501.3
1.0	514.9	609.0	549.3	590.1	538.0	434.9
3.0	623.7	737.7	665.4	714.8	651.7	526.8

**Table 5 polymers-15-00707-t005:** Cohesion at 20 and 70 °C of Q2-7358 resin-based adhesives with clinoptilolite.

Filler Content [wt%]	Cohesion [h]					
	K 0.00	K 0.01	K 0.10	K 0.50	K 1.00	K 2.00
20 °C						
0.1	24.6	20.1	40.3	45.1	>72	>72
0.5	>72	>72	>72	>72	>72	>72
1.0	>72	>72	>72	>72	>72	>72
3.0	>72	>72	>72	>72	>72	>72
70 °C						
0.1	0.5	0.8	1.3	1.5	14.2	20.4
0.5	>72	>72	>72	>72	>72	>72
1.0	>72	>72	>72	>72	>72	>72
3.0	14.2	9.2	10.4	56.2	>72	51.0

**Table 6 polymers-15-00707-t006:** Cohesion at 20 and 70 °C of Q2-7355 resin-based adhesives with clinoptilolite.

Filler Content [wt%]	Cohesion [h]					
	K 0.00	K 0.01	K 0.10	K 0.50	K 1.00	K 2.00
20 °C						
0.1	>72	>72	>72	>72	>72	>72
0.5	>72	>72	>72	>72	>72	>72
1.0	>72	>72	>72	>72	>72	>72
3.0	>72	>72	>72	>72	>72	>72
70 °C						
0.1	>72	>72	>72	>72	>72	>72
0.5	>72	>72	>72	>72	>72	>72
1.0	>72	>72	>72	>72	>72	>72
3.0	>72	>72	>72	>72	>72	8.7

**Table 7 polymers-15-00707-t007:** The maximal working temperature of silicon adhesives with clinoptilolite.

Filler Content [wt%]	SAFT [°C]					
	K 0.00	K 0.01	K 0.10	K 0.50	K 1.00	K 2.00
Q2-7358						
0.1	89	81	98	95	166	150
0.5	217	>225	>225	>225	210	211
1.0	163	210	201	198	200	187
3.0	159	201	200	197	166	148
Q2-7355						
0.1	>225	219	219	217	217	216
0.5	219	>225	>225	210	210	210
1.0	213	219	219	211	211	210
3.0	210	211	217	217	206	209

**Table 8 polymers-15-00707-t008:** Shrinkage of Q2-7358 resin-based pressure-sensitive adhesives with clinoptilolite.

**K 0.00**												
**Shrinkage (%)**												
**Filler Content** **[wt%]**	**10 min**	**30 min**	**1 h**	**3 h**	**8 h**	**24 h**	**2 Days**	**3 Days**	**4 Days**	**5 Days**	**6 Days**	**7 Days**
0.1	0.200	0.240	0.270	0.300	0.320	0.360	0.400	0.450	0.470	0.490	0.510	0.510
0.5	0.170	0.203	0.232	0.289	0.299	0.316	0.395	0.400	0.437	0.437	0.437	0.437
1.0	0.087	0.295	0.324	0.379	0.398	0.413	0.452	0.452	0.452	0.452	0.452	0.452
3.0	0.136	0.225	0.227	0.267	0.286	0.306	0.346	0.367	0.380	0.380	0.380	0.380
**K 0.01**												
**Shrinkage (%)**												
**Filler content** **[wt%]**	**10 min**	**30 min**	**1 h**	**3 h**	**8 h**	**24 h**	**2 Days**	**3 Days**	**4 Days**	**5 Days**	**6 Days**	**7 Days**
0.1	0.180	0.240	0.280	0.300	0.340	0.370	0.400	0.430	0.440	0.450	0.470	0.470
0.5	0.089	0.202	0.231	0.289	0.208	0.317	0.394	0.410	0.440	0.442	0.442	0.442
1.0	0.087	0.300	0.326	0.370	0.398	0.400	0.470	0.473	0.473	0.473	0.473	0.473
3.0	0.110	0.211	0.221	0.232	0.254	0.298	0.326	0.367	0.380	0.380	0.380	0.380
**K 0.10**												
**Shrinkage (%)**												
**Filler content** **[wt%]**	**10 min**	**30 min**	**1 h**	**3 h**	**8 h**	**24 h**	**2 Days**	**3 Days**	**4 Days**	**5 Days**	**6 Days**	**7 Days**
0.1	0.098	0.239	0.270	0.334	0.376	0.399	0.518	0.536	0.536	0.536	0.536	0.536
0.5	0.085	0.208	0.235	0.290	0.327	0.347	0.450	0.466	0.466	0.466	0.466	0.466
1.0	0.090	0.194	0.210	0.224	0.288	0.311	0.334	0.354	0.366	0.372	0.399	0.399
3.0	0.099	0.162	0.220	0.276	0.322	0.387	0.389	0.407	0.428	0.428	0.428	0.428
**K 0.50**												
**Shrinkage (%)**												
**Filler content (pph)**	**10 min**	**30 min**	**1 h**	**3 h**	**8 h**	**24 h**	**2 Days**	**3 Days**	**4 Days**	**5 Days**	**6 Days**	**7 Days**
0.1	0.109	0.251	0.282	0.345	0.388	0.411	0.529	0.547	0.547	0.547	0.547	0.547
0.5	0.095	0.218	0.245	0.300	0.337	0.357	0.460	0.476	0.476	0.476	0.476	0.476
1.0	0.100	0.204	0.220	0.234	0.298	0.321	0.344	0.364	0.376	0.382	0.409	0.409
3.0	0.109	0.172	0.230	0.286	0.332	0.397	0.399	0.417	0.438	0.438	0.438	0.438
**K 1.00**												
**Shrinkage (%)**												
**Filler content** **[wt%]**	**10 min**	**30 min**	**1 h**	**3 h**	**8 h**	**24 h**	**2 Days**	**3 Days**	**4 Days**	**5 Days**	**6 Days**	**7 Days**
0.1	0.120	0.261	0.292	0.355	0.398	0.421	0.539	0.558	0.558	0.558	0.558	0.558
0.5	0.104	0.227	0.254	0.309	0.346	0.366	0.469	0.485	0.485	0.485	0.485	0.485
1.0	0.090	0.194	0.210	0.224	0.288	0.311	0.334	0.354	0.366	0.372	0.399	0.399
3.0	0.119	0.182	0.240	0.296	0.342	0.407	0.409	0.427	0.448	0.448	0.448	0.448
**K 2.00**												
**Shrinkage (%)**												
**Filler content** **[wt%]**	**10 min**	**30 min**	**1 h**	**3 h**	**8 h**	**24 h**	**2 Days**	**3 Days**	**4 Days**	**5 Days**	**6 Days**	**7 Days**
0.1	0.121	0.274	0.305	0.368	0.411	0.434	0.552	0.570	0.570	0.570	0.570	0.570
0.5	0.105	0.238	0.265	0.320	0.357	0.377	0.480	0.496	0.496	0.496	0.496	0.496
1.0	0.085	0.189	0.205	0.219	0.283	0.306	0.329	0.349	0.361	0.367	0.394	0.394
3.0	0.089	0.152	0.210	0.266	0.312	0.377	0.379	0.397	0.418	0.418	0.418	0.418

**Table 9 polymers-15-00707-t009:** Shrinkage of Q2-7355 resin-based pressure-sensitive adhesives with clinoptilolite.

**K 0.00**												
**Shrinkage (%)**												
**Filler Content** **[wt%]**	**10 min**	**30 min**	**1 h**	**3 h**	**8 h**	**24 h**	**2 Days**	**3 Days**	**4 Days**	**5 Days**	**6 Days**	**7 Days**
0.1	0.407	0.413	0.477	0.512	0.557	0.614	0.641	0.697	0.713	0.754	0.754	0.754
0.5	0.307	0.311	0.324	0.379	0.418	0.449	0.486	0.513	0.584	0.631	0.669	0.669
1.0	0.065	0.069	0.072	0.075	0.078	0.082	0.089	0.091	0.097	0.104	0.112	0.131
3.0	0.050	0.054	0.056	0.059	0.061	0.068	0.071	0.077	0.081	0.086	0.092	0.097
**K 0.01**												
**Shrinkage (%)**												
**Filler content** **[wt%])**	**10 min**	**30 min**	**1 h**	**3 h**	**8 h**	**24 h**	**2 Days**	**3 Days**	**4 Days**	**5 Days**	**6 Days**	**7 Days**
0.1	0.418	0.424	0.488	0.523	0.568	0.625	0.652	0.708	0.724	0.765	0.765	0.765
0.5	0.318	0.322	0.335	0.390	0.429	0.460	0.497	0.524	0.595	0.642	0.680	0.680
1.0	0.076	0.080	0.083	0.086	0.089	0.093	0.100	0.102	0.108	0.115	0.123	0.142
3.0	0.061	0.065	0.067	0.070	0.072	0.079	0.082	0.088	0.092	0.097	0.103	0.108
**K 0.10**												
**Shrinkage (%)**												
**Filler content (pph)**	**10 min**	**30 min**	**1 h**	**3 h**	**8 h**	**24 h**	**2 Days**	**3 Days**	**4 Days**	**5 Days**	**6 Days**	**7 Days**
0.1	0.411	0.417	0.481	0.516	0.561	0.618	0.645	0.701	0.717	0.758	0.758	0.758
0.5	0.311	0.315	0.328	0.383	0.422	0.453	0.490	0.517	0.588	0.635	0.673	0.673
1.0	0.069	0.073	0.076	0.079	0.082	0.086	0.093	0.095	0.101	0.108	0.116	0.135
3.0	0.054	0.058	0.060	0.063	0.065	0.072	0.075	0.081	0.085	0.090	0.096	0.101
**K 0.50**												
**Shrinkage (%)**												
**Filler content** **[wt%]**	**10 min**	**30 min**	**1 h**	**3 h**	**8 h**	**24 h**	**2 Days**	**3 Days**	**4 Days**	**5 Days**	**6 Days**	**7 Days**
0.1	0.418	0.424	0.488	0.523	0.568	0.625	0.652	0.708	0.724	0.765	0.765	0.765
0.5	0.318	0.322	0.335	0.390	0.429	0.460	0.497	0.524	0.595	0.642	0.680	0.680
1.0	0.076	0.080	0.083	0.086	0.089	0.093	0.100	0.102	0.108	0.115	0.123	0.142
3.0	0.061	0.065	0.067	0.070	0.072	0.079	0.082	0.088	0.092	0.097	0.103	0.108
**K 1.00**												
**Shrinkage (%)**												
**Filler content** **[wt%]**	**10 min**	**30 min**	**1 h**	**3 h**	**8 h**	**24 h**	**2 Days**	**3 Days**	**4 Days**	**5 Days**	**6 Days**	**7 Days**
0.1	0.293	0.310	0.370	0.396	0.430	0.491	0.512	0.603	0.613	0.705	0.705	0.705
0.5	0.222	0.250	0.273	0.311	0.390	0.411	0.496	0.550	0.610	0.647	0.647	0.647
1.0	0.077	0.080	0.084	0.090	0.095	0.096	0.100	0.105	0.112	0.134	0.156	0.156
3.0	0.043	0.050	0.052	0.054	0.061	0.063	0.069	0.073	0.077	0.083	0.090	0.096
**K 2.00**												
**Shrinkage (%)**												
**Filler content** **[wt%]**	**10 min**	**30 min**	**1 h**	**3 h**	**8 h**	**24 h**	**2 Days**	**3 Days**	**4 Days**	**5 Days**	**6 Days**	**7 Days**
0.1	0.223	0.265	0.283	0.330	0.397	0.418	0.461	0.532	0.610	0.647	0.647	0.647
0.5	0.210	0.242	0.250	0.283	0.278	0.311	0.333	0.353	0.377	0.417	0.471	0.471
1.0	0.019	0.120	0.142	0.156	0.196	0.217	0.255	0.306	0.313	0.339	0.348	0.348
3.0	0.039	0.041	0.044	0.046	0.050	0.055	0.057	0.060	0.063	0.066	0.069	0.075

## Data Availability

Not applicable.

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
