# Peer review of "Influence of Acid-Modified Clinoptilolite on the Self-Adhesive Properties of Silicone Pressure-Sensitive Adhesives"

_polymers, 2023, doi:10.3390/polym15030707_

Round 1
Reviewer 1 Report
I have uploaded a word file.

Author Response
The authors would like to thank to the Reviewer and truly appreciate his comments, questions and corrections. Please find the detailed answers below.
In this work, heat-resistant self-adhesive silicone tapes were obtained via addition of acid-modified filler, clinoptilolite. Various experimental methods were employed to characterize the multiple properties of the new silicone PSAs. However, there is still much room for improvement in both writing and logic. And the following questions should be solved prior to be considered to be published.
- Lines 28-31: “Pressure-sensitive adhesives (PSA) are defined as a group of adhesives exhibiting significant adhesive strength and stickiness with the substrate at room temperature. Importantly, good adhesive parameters are not the result of chemical reaction between the adhesive and the substrate [2–4].” Here, I think the key property of pressure-sensitive should be given an introduction or a definition.
We made a change to the manuscript, we hope now it is correct.
- Lines 32-33: The sentence “The most commonly used are solvent-borne PSA.” Needs an appropriate reference.
We have corrected the manuscript.
- Line 34: “They are commonly used to for……”, here the word “to” seems redundant.
We have corrected the manuscript.
- Line 39: “such as e.g.” the expressions are repeated.
We have corrected the manuscript.
- Line 44: “is” should be replaced by “are”.
We have corrected the manuscript.
- Line 49: “range” should be “ranges”.
We have corrected the manuscript.
- Line 50: “an organic peroxide” should be “organic peroxides”.
We have corrected the manuscript.
- Lines 56-58: “Silicone pressure-sensitive adhesives are widely used in adhesive tapes and labels when application conditions or the nature of the substrate surface exceed the performance limits of organic PSA adhesives.” First, “organic PSA adhesives” should be “organic PSAs”. Second, what are silicone pressure-sensitive adhesives? And what is the different between silicone pressure-sensitive adhesives and organic PSAs?
We made a change to the manuscript, we hope now it is correct.
- Lines 61-71: The background information is weak. How come the clinoptilolite was introduced into the silicone PSAs? And why use acid to treat clinoptilolite? More background information should be provided and elucidated. And what are 10 % Н4EDTA/Kl, HKl-1, 10 % Н4EDTA/НKl-1, and 10 % PW12-HPA/НKl-1?
We have corrected the manuscript.
We have added the meaning of the symbols quoted.
- Line 82, when an abbreviation firstly appeared, its full name should be provided.
We have corrected the manuscript.
Finally, we hope that corrections made in the manuscript fulfill reviewer suggestions and allow editor to make positive decision about acceptation of our contribution for publishing in this journal.
With regards,
Adrian Krzysztof Antosik
Marlena Musik
Piotr Miądlicki
Mateusz Weisbrodt
Katarzyna Wilpiszewska
Reviewer 2 Report
In the present study, the authors presented the preparation of a new "eternally alive adhesive" based on silicone pressure-sensitive adhesives with clinoptilolite. To obtain suitable fillers, clinoptilolite was treated with sulphuric acid (VI) solutions at various molar concentrations. New self-adhesive tapes were obtained. The background of the work is interesting. The topic falls within the target journal.
The logic of the work is good, and the language is acceptable. I got through the whole article, and found besides some minor points, it can be considered for publication after some modifications.
1. In the paper, the pressure-sensitive adhesives is called “PSA”. In my opinion, it is better cited as “PSAs”. Please check the following names.
2. The authors mentioned the hydrophobicity of the solid surface. Actually, besides the surface energy, it is closely related with the roughness of the surfaced. Please see the related work:
Liu et al., A new look on wetting models: continuum analysis. Science China Physics, Mechanics & Astronomy, 2012, 55(11): 1–9.
3. There are so many experiments and there is a lack of though theoretical analysis. As a suggestion, the molecular simulation can be made, at least in the near future. This can be noted.
4. In the work the authors mentioned the adhesion. Actually the adhesion is formed between two interfaces is due to the work of adhesion, or the interfacial energy. This is a very important parameter to characterize the adhesion behavior. The authors at least should make a note on this issue. See the related work: Liu et al., A unified analysis of a micro-beam, droplet and CNT ring adhered on a substrate: Calculation of variation with movable boundaries. Acta Mechanica Sinica, 2013.
5. In L155, the expression “H2O” should be revised.
6. In table 2, what is the unit of the adhesion? Is it “Adhesion [N/25 mm]”?
7. The format of the references should be unified and they should be retyped.
Author Response
The authors would like to thank to the Reviewer and truly appreciate his comments, questions and corrections. Please find the detailed answers below.
In the present study, the authors presented the preparation of a new "eternally alive adhesive" based on silicone pressure-sensitive adhesives with clinoptilolite. To obtain suitable fillers, clinoptilolite was treated with sulphuric acid (VI) solutions at various molar concentrations. New self-adhesive tapes were obtained. The background of the work is interesting. The topic falls within the target journal.
The logic of the work is good, and the language is acceptable. I got through the whole article, and found besides some minor points, it can be considered for publication after some modifications.
- In the paper, the pressure-sensitive adhesives is called “PSA”. In my opinion, it is better cited as “PSAs”. Please check the following names.
We agree with the reviewer and have introduced changes using the abbreviation PSA in the singular and PSAs in the plural.
- The authors mentioned the hydrophobicity of the solid surface. Actually, besides the surface energy, it is closely related with the roughness of the surfaced. Please see the related work:
Liu et al., A new look on wetting models: continuum analysis. Science China Physics, Mechanics & Astronomy, 2012, 55(11): 1–9.
We made a suggestion reviewer.
- There are so many experiments and there is a lack of though theoretical analysis. As a suggestion, the molecular simulation can be made, at least in the near future. This can be noted.
Thank you for the interesting suggestion, I will gladly use it for new publications to better direct the conduct of empirical research.
- In the work the authors mentioned the adhesion. Actually the adhesion is formed between two interfaces is due to the work of adhesion, or the interfacial energy. This is a very important parameter to characterize the adhesion behavior. The authors at least should make a note on this issue. See the related work: Liu et al., A unified analysis of a micro-beam, droplet and CNT ring adhered on a substrate: Calculation of variation with movable boundaries. Acta Mechanica Sinica, 2013.
Thank you for suggesting the article, we reviewed it and used it to enrich the introduction.
- In L155, the expression “H2O” should be revised.
We have corrected the manuscript.
- In table 2, what is the unit of the adhesion? Is it “Adhesion [N/25 mm]”?
The unit of adhesion is N/25mm, we have improved the visibility in the table, we hope it is now clearly visible.
- The format of the references should be unified and they should be retyped.
We made a change to the manuscript, we hope now it is correct.
Finally, we hope that corrections made in the manuscript fulfill reviewer suggestions and allow editor to make positive decision about acceptation of our contribution for publishing in this journal.
With regards,
Adrian Krzysztof Antosik
Marlena Musik
Piotr Miądlicki
Mateusz Weisbrodt
Katarzyna Wilpiszewska